# Total Mercury Content in the Tissues of Freshwater Chelonium (*Podocnemis expansa*) and a Human Health Risk Assessment for the Amazon Population in Brazil

**DOI:** 10.3390/ijerph20156489

**Published:** 2023-08-01

**Authors:** Fábio Júnior Targino, Joanna Damazio de Nunes Ribeiro, Julia Siqueira Simões, Carla Silva Carneiro, Stella Maris Lazzarini, Aline Ramos Souza, Micheli da Silva Ferreira, Sergio Borges Mano, Eliane Teixeira Mársico

**Affiliations:** 1Faculty of Veterinary, Universidade Federal Fluminense (UFF), Niterói 24230-321, Brazil; fabiotargino@id.uff.br (F.J.T.); joannadnr@gmail.com (J.D.d.N.R.); julia_simoes@hotmail.com (J.S.S.); micheliferreira@id.uff.br (M.d.S.F.); sergiomano@id.uff.br (S.B.M.); 2Faculty of Pharmacy, Universidade Federal do Rio de Janeiro (UFRJ), Rio de Janeiro 21941-580, Brazil; carlasilvacarneiro@gmail.com; 3Centro de Pesquisa e Preservação de Mamíferos Aquáticos (CPPMA), Eletronorte, Presidente Figueiredo 69736-000, Brazil; stellamarislazz@gmail.com (S.M.L.); alineramos.sza@gmail.com (A.R.S.); 4Institute of Biological Sciences, Universidade Federal do Amazonas (UFAM), Manaus 69077-000, Brazil

**Keywords:** public health, food safety, environmental pollution, trace elements, biomonitoring, freshwater toxicology

## Abstract

Researchers recognize the silent, negative and deleterious effects caused by mercury pollution in gold mining areas. Freshwater turtles are culturally part of the diet of riverside populations in the Amazon region and this area presents mercury (Hg) pollution issues mainly due to gold mining activities. Thus, this research aimed to evaluate the total mercury (THg) content in the different organs of Amazonian giant river turtle (*Podocnemis expansa*) and carry out a human health risk assessment associated with the consumption of these animals. This study was conducted in the Vila Balbina, municipality of Presidente Figueiredo, state of Amazonas, Brazil. Skin (*n* = 28), muscle (*n* = 19) and brain (*n* = 2) samples were analyzed by Atomic Absorption Spectrometry (TDA-AAS) and a DMA-80™ mercury analyzer was used for the total mercury determinations. The average values found for THg in the skin, muscle and brain samples were, respectively, 0.1045 mg·kg^−1^, 0.1092 mg·kg^−1^ and 0.0601 mg·kg^−1^. Thus, THg was observed even though the *P. expansa* were kept in captivity, possibly due to previous contamination by air, water and food. The Hazard Quotient (HQ) was calculated considering a 9.07 g·day^−1^ intake dose of *P. expansa* and the consumption of turtles once a week showed an HQ = 2.45, which may cause long-term injuries to human health. Although the muscle concentrations were below the maximum limit established by the World Health Organization (WHO) and Brazilian regulatory agencies, it is important to evaluate consumption factors such as amount ingested, frequency and animal gender, which may cause a potential risk to regular consumers due to mercury bioaccumulation. The WHO may consider various aspects in order to warn the Amazon population about the severity and silent hazard of this metal, especially due to the importance of this matrix in the region. This region urgently needs government actions to inhibit clandestine mining and to prevent future serious, chronic health problems of the entire population.

## 1. Introduction

Mercury (Hg) is a toxic trace element found in the Amazonian ecosystem due to its use in the amalgamation stage of gold extraction [1,2,3]. This toxic contaminant requires public health attention due to its toxicity, high absorption and low excretion after ingestion [4]. The toxic element is present in the aquatic environment where it is methylated, bioaccumulates and biomagnifies, providing the possibility for incorporation by human beings, who are at the top of the food chain [5]. Metal pollution, although less perceptible compared to other forms of pollution or contamination, can have profound and far-reaching impacts on ecosystems and human health [6]. Researchers have conducted extensive studies on mercury contamination in water, sediment and aquatic fauna across various countries and regions, including China [7], India [8], Italy [9] and Japan [10], as well as the Peruvian [11] and Brazilian Amazon [2].

Mercury can have toxic effects on tissues and systems such as the nervous, digestive and immune systems, and organs such as the lungs, kidneys, skin and eyes [12]. The organic form, methylmercury (MeHg), is a potent neurotoxin inducing oxidative stress activity, neuroinflammatory action and changes in heart rate [13]. Mercury is eliminated into the environment in the inorganic form and may undergo sediment methylation by physical, chemical and biological mechanisms, turning into MeHg, the most toxic form of this metal [14,15]. Due to its geochemical cycle, mercury can be found in areas far away from its mobilized source [16]. Inorganic Hg reaches the atmosphere, and, by precipitation, is deposited on the bottom of rivers, lakes and other watercourses [17]. Furthermore, humans may demonstrate pathognomonic effects [18].

To protect human health and limit dietary exposure to toxic metals, international agencies and Brazilian regulations have provided guidelines for the human intake of trace elements [19,20,21]. According to the World Health Organization (WHO), to estimate the amount of mercury per unit of body weight that can be ingested over a lifetime without health risks, the joint FAO/WHO Expert Committee recommended Provisional Tolerable Weekly Intakes (PTWIs) for MeHg of 1.6 mg·kg^−1^ body weight [22].

In recent years, particularly from 2016 to 2022, there has been a significant increase in illegal mining activity in the Brazilian Amazon, with a focus on indigenous territories. According to data from the Instituto Nacional de Pesquisa Espacial (INPE), there has been a staggering 787% rise in illegal mining activity within indigenous territories during this period [23]. This surge can be attributed to various factors, including the encouragement of such activities, inadequate monitoring and enforcement measures and the weakening of environmental regulations in the country [24,25]. Economic crisis and unemployment during the COVID-19 pandemic forced thousands of people to work in illegal mining in the Amazon [25].

The Amazon population’s diet varies according to the season, but aquatic animals are the primary protein source for most riverside people, especially children. The consumption of turtles by natives and settlers is historically and culturally developed by Amazon communities [26], and the Amazonian giant river turtle (*Podocnemis expansa)* is the most consumed species. After colonization, chelonian commerce improved and the local population could explore the meat, eggs, oil and carapace as an additional financial source. Due to their long lifespan, aquatic turtles can accumulate Hg over extended periods and therefore it could be of interest to monitor environmental contamination [26,27]. The history of gold mining and hydroelectric power plants has influenced Hg availability in the environment and living organisms [5,28,29].

Furthermore, the consumption of chelonium meat from illegal hunting can lead to environmental imbalance, given the important ecological role turtles play in the Amazon’s biodiversity [30]. Additionally, consumers are at risk of exposure to infectious agents, such as Enterobacteriaceae, notably *Salmonella* sp. [31,32]. Therefore, it is essential to develop research and environmental biomonitoring programs to measure the Hg concentration in *P. expansa* and assess the risk of exposure to this population. Thus, this research aimed to evaluate the presence of total mercury (THg) in different organs of the *P. expansa* chelonian species and carry out a human health risk assessment associated with the consumption of these animals.

## 2. Materials and Methods

### 2.1. Experimental Area

The location studied was Vila Balbina, in the municipality of Presidente Figueiredo, located in the North of Amazonas State (latitude 1°55′07.0 S and longitude 59°28′05.9 W), Brazil. This location was considered for study due to the extensive mining activity that has occurred there for decades. The *P. expansa* samples (Figure 1) were obtained from the Centro de Preservação e Pesquisa de Quelônios Aquáticos (CPPQA) (Figure 2) located near the River Uatumã mouth, a tributary of the black waters inserted in an area with anthropogenic actions such as metal extractions (niobium-Nb and tin-Sn).

### 2.2. Chelonian Sampling

Twenty-eight samples of *P. expansa* were collected in June 2018 from the CPPQA captivity area, and the carapace length of each animal was measured. The evaluated animals (Figure 1) lived in confinement, kept in captivity at the CPPQA. These animals consisted of breeding males and female matrices, which are part of the restocking program on the beaches of the Uatumã river. Despite being kept in captivity, the food supply and water used in their enclosures and macrophyte production tanks came from the Uatumã river, downstream from the Balbina Hydroelectric Power Plant.

Skin and muscle samples (~2 cm^3^) were obtained by biopsy punch (3 mm) from the pelvic member using a previous local anesthesia. Considering the difficulty of animal handling, 28 skin samples and 19 muscle samples were obtained, but only two brain samples were collected by the CPPQA when the animals died.

The animals underwent a biopsy using a punch, and they received subsequent treatment from veterinary doctors and research veterinarians at CPPQA. The treatment involved applying povidone-iodine and a healing ointment containing gentamicin, sulfanilamide, urea, vitamin A and excipients. This treatment regimen was administered daily for seven days to the wound area. The animals were housed in their original enclosure, which was in a designated area with net protection. This protocol was approved by the Comitê de Ética na Utilização de Animais (CEUA) of Universidade Federal Fluminense (UFF) under protocol number: 1012300319.

The samples were frozen (−18 °C ± 2 °C), stored in polypropylene packing and flown to a chemical laboratory for analysis at the Federal Fluminense University. The animals were gender-identified by observation of the reproductive organ dimorphism [33].

### 2.3. Chelonian Consumption Data

Ninety-one Balbina residents answered a questionnaire about the consumption of chelonians in the area. A semi-structured questionnaire was administered in an individual interview format. The inclusion criteria required participants to be over eighteen years of age and residents of Vila Balbina, located in the municipality of Presidente Figueiredo. The questionnaire consisted of two sections: (1). Socioeconomic factors (including gender, age, occupation, history of involvement in mining activities and education level); (2). Chelonian consumption patterns (including consumption of chelonian meat and turtle eggs, preferred species for consumption, frequency of consumption and annual amount of consumption of chelonian meat and eggs).

### 2.4. Instrumentation

The Direct mercury analyzer DMA80™ (Milestone Inc., Shelton, CT, USA) involves thermal decomposition, catalytic conversion, amalgamation and atomic absorption spectrophotometry. The heating stages are implemented to first dry and then thermally decompose the sample in a quartz cuvette, and a flow of oxygen carries the decomposition products through a hot catalyst bed. The mercury content is reduced to Hg (0) and then carried along with the reaction gases to an amalgamator, where the mercury is selectively trapped. Subsequently, the amalgamator is heated and releases the trapped mercury to the single beam, fixed wavelength atomic absorption spectrophotometer. The absorbance is measured at 253.7 nm as a function of the mercury content. All non-mercury vapors and decomposition products are flushed from the system by the continuous flow of gas (Milestone Inc., Shelton, CT, USA).

### 2.5. Analytical Procedures

The analyses were carried out according to the analytical procedures of the US Environmental Protection Agency (US EPA) 7473 (2007) [34]. An aliquot of 0.210 g of the chelonian sample was directly and accurately weighed into a quartz cuvette. The DMA80™ used does not require pretreated samples. According to the WHO/FAO recommendations [22], the samples were weighed on a wet weight basis (wwb).

The samples were submitted to the following heating stages: starting temperature of 200 ± 10 °C; drying stage for 60 s at 250 ± 10 °C; thermal decomposition at 650 ± 10 °C for 120 s and finally 100 s for the analytical reading and cooling. An amalgamator selectively trapped the Hg, and the oxygen flow removed any remaining gases or decomposition products (24 s); the amalgamator was rapidly heated (12 s), releasing the Hg vapor. The atomic absorbance was read at 253.7 nm.

The accuracy of the analytical procedure was evaluated by analyzing the Certified Reference Material (CRM) of DORM-2 (dogfish muscle tissue) obtained from the National Research Council Canada (NRCC). The CRM control samples were analyzed without any treatment, and non-analyzed aliquots were used to determine the moisture content according to the certificate instructions for analysis. The certified material concentration was determined in triplicate and reached 4.470 µg·g^−1^. The mean recovery was 94%.

### 2.6. Limits of Detection and Quantification

The limits of detection (LOD) and quantification (LOQ) were calculated from the mean concentration and relative standard deviation of ten blanks and the slope of the analytical curve (1.5 × 10^−6^ mg·kg^−1^ of THg). These limits were calculated as 3 and 10 times the standard deviation of 10 blanks and multiplied by the dilution factor used for sample preparation (1 g of sample/25 mL) using the following equations:LOD = 3 × s and LOQ = 10 × s
where: s = the average standard deviation in units of instrumental response, of at least ten independent determinations of analytical blanks.

The limits of detection (LOD) and quantification (LOQ) were 94 µg·kg^−1^ and 314 µg·kg^−1^, respectively. The recovery value mean was 84.93%. The value of the LOQ was confirmed using ten independent replicates.

### 2.7. Human Health Risk Assessment

The human health risk was assessed by the US EPA/WHO method (1989) [35], assessing the exposure and characterizing the risk associated with the consumption of *P. expansa*. The exposure assessment was represented by I (Intake dose), which involves the description of the nature and size of the populations studied exposed to the chemical agent (Hg in this study), and the magnitude and duration of the exposure, calculated according to the following equation:I = (CF × IR × FI × EF × ED)/(BW × AT)

The Intake dose (I) was defined by the ratio between the sample and the estimated weight of the exposed population. The following factors were taken into account: the chemical concentration (CF) in *P. expansa* muscle (male, female and total mean); the intake rate (IR), taken as the average daily consumption of turtle of 9.07 g·person·day^−1^ as estimated by Isaac et al. (2015) [36]; the fraction of food ingested (FI) obtained from a contaminated source (considered as the total food fraction in this study) [35,37]; the exposure frequency (EF) consumption considered annually, monthly and weekly; the exposure duration (ED), established from the average life expectancy for Brazilian people of 75.8 years [38]; body weight (BW), considered as 70 kg [35]; and average time (AT) as the annual exposure frequency by individuals during their life expectancy (AT = ED × 365 days).

For the risk characterization, the intake dose (I) was compared with the RfD (Reference dose of Hg), which, according to US EPA (2011) [20], is 3.0 × 10^−4^ mg·kg·day^−1^, calculated using the following equation.
HQ = I/RfD

HQ greater than 1 (HQ > 1) order of magnitude indicating health risks.

### 2.8. Statistical Analysis

The normality of the data was confirmed with a Shapiro–Wilk test (*p* > 0.05). The one-way ANOVA and Tukey’s test were applied using XLSTAT™ 2014.1 (Addinsoft Inc., New York, NY, USA) with a significance level of 95% (*p* < 0.05) to establish any significant differences between the means. A correlation analysis was applied, and Pearson’s linear coefficient was calculated amongst the muscle and skin tissues. All analyses were carried out in triplicate.

## 3. Results and Discussion

### 3.1. Results

#### 3.1.1. Total Mercury (THg) Content

The average carapace length for females was 74.42 cm (*n* = 14), and it was 47.18 cm (*n* = 16) for males. The present results show that although the female carapaces were larger than the males, higher THg concentrations were not observed in the female animals. The muscle tissue samples showed the highest concentrations of THg (0.6929 mg·kg^−1^) as compared to the maximum values in the skin of (0.5457 mg·kg^−1^) and in the brain of (0.0610 mg·kg^−1^). The mean THg concentrations found in the skin (*n* = 28), muscle (*n* = 19) and brain (*n* = 2) were, respectively, 0.1045 mg·kg^−1^, 0.1092 mg·kg^−1^ and 0.0601 mg·kg^−1^. The results are shown in Table 1.

Regarding the tissues, the muscle should be considered the most important because of its higher accumulation of THg in *P. expansa* compared to the skin, but the present research found no significant difference (*p* < 0.05). However, a notable difference was detected in the tissue concentrations, considering a minimum of “not detected” in the muscle and skin and a maximum of 0.6929 mg·kg^−1^ in the muscle. The average values found in the male samples demonstrated greater THg bioaccumulation in the male muscle than in the female muscle, and a significant difference (*p* < 0.05) was observed between the female muscle samples and the samples of the other tissues.

A positive correlation, regardless of gender, was observed between the skin and muscle THg concentrations (r = 0.87). The present research did not find a correlation between carapace length and muscle or skin THg concentrations for males (r = 0.011) or of females (r = 0.15), possibly due to the high variation between the tissue concentrations in the animal diet and the diversity of their origins. Furthermore, regarding the coefficient of determination (R^2^), which measures the proportion in which the variables are related, a slight difference was observed when comparing tissues (skin and muscles) and sex, with R^2^ = 0.50 for males and 0.43 for females.

#### 3.1.2. Human Health Risk Assessment

Amongst those interviewed, 18% (15/91) consumed chelonian meat, especially tracajá (*Podocnemis unifilis*) and *P. expansa*. However, only 8% (7/91) reported consuming the eggs. Regarding the consumption data used to calculate the HQ, 8/91 ate chelonian meat at least once a year, 5/91 ate it once a month, and 2/91 ate it once a week during the egg-laying months (September to December). The people interviewed confirmed that these animals and eggs were caught in Vila Balbina, Presidente Figueiredo, Amazonas, Brazil.

The Hazard Quotient (HQ) was calculated for the different consumption intervals (annually, monthly and weekly) with respect to the THg concentrations in male plus female *P. expansa* and the mean THg values for both genders separately. Table 2 shows the HQ results obtained for the human risk assessment of *P. expansa* muscle.

### 3.2. Discussion

#### 3.2.1. THg Content

The total mercury content in the different turtle tissues is related to Hg kinetics. Researchers have indicated that the total mercury content in freshwater fish is related to the species and habitat evaluated [5,39,40]. Brazilian studies have reported a THg content in Amazon fish muscle of over 0.500 mg·kg^−1^, above the limit recommended by World Health Organization [19,41,42]. Although there are few studies with Amazonian freshwater chelonians, it is essential to consider the influence of the environment on the contents of trace elements.

The *P. unifilis* caught in the lower River Xingu accumulated lower concentrations of THg throughout its life when compared to other species of Amazon turtles. Thus, the Amazon basin rivers have mercury in a bioavailable form that organisms can accumulate. Recent studies have shown that *P*. *unifilis* and *P. expansa* are predominantly herbivorous, but may consume insects, crustaceans and mollusks [43,44].

The present authors observed that some studies indicate a difference in feeding preference between genders, with the females consuming more fruits and seeds and the males more stalks and shoots [45]. Thus, it was postulated that the lack of significant difference in Hg concentrations between males and females in this study could be attributed to the absence of variation in trophic levels among species gender [46]. Due to the maintenance of the animals analyzed in this research in captivity, the diet of males and females is similar, consisting of offering vegetables, fruits and macrophytes grown in the CPPQA itself. It assumes that in free-living animals at the mouth of the Uatumã river, the levels of Hg can be significantly different in males and females, due to eating habits, the period of prolonged fasting that females have before the laying phase and the transfer of inorganic chemical trace elements to eggs. Research has identified a maternal transfer of metals (essential or not) to the eggs in tests with Testudines nesting eggs, triggering genotoxic effects in the hatchlings [47].

Research carried out in the Amazon evaluated non-lethal and non-invasive ways of collecting samples from Amazonian chelonian species for Hg monitoring [48]. They found no significant differences between the carapace and internal organs of *Podocnemis erythrocephala* and *Podocnemis sextuberculata*.

Research conducted in a sustainable development reserve in Amazonas, Brazil evaluated THg concentration in *P. unifilis*. The study found an average THg concentration mean of 0.011 mg·kg^−1^ in six muscle samples [49].

Different THg concentrations may be observed between males and females due to changes in eating habits [50] and the elimination of Hg via the eggs during the reproductive period [51]. However, in contrast, the present study diverged from this observation and the THg concentration for males did not significantly differ from that of the females. This study found 0.0624 mg·kg^−1^ THg in *P. expansa* muscle samples, 0.432 mg·kg^−1^ THg in *Chelus fimbriatus* muscle and 0.033 mg·kg^−1^ THg in *P. erythrocephala* muscle [50,52]. These animals present distinct eating behaviors between the genders and life periods. Small amounts of fish and crab were observed in studies that analyzed the chelonian gastrointestinal content, but their primary food source is seeds and fruits, especially those from riverside trees and flooded forests. However, the literature also reported that juvenile *P. expansa*, in captivity, prefer to eat fish, insects and other meat [52,53].

#### 3.2.2. Mechanism for THg Distribution

Hg is a metal that occurs naturally in the Amazonian environment in small quantities but constitutes an important pollutant derived from anthropogenic activities [40,54]. Once released into the environment, Hg is oxidized and will be available to be methylated, producing methylmercury (MeHg), a neurotoxic and teratogenic substance. MeHg undergoes bioaccumulation in aquatic organisms and is biomagnified through the trophic chain, constituting a source of exposure to man via the food chain, mainly due to fish consumption [18]. Traditional populations of fishermen, indigenous and riverside, are more exposed to this type of contamination due to the higher frequency of fish intake [55].

Elemental Hg in reptiles is well-absorbed by inhalation through the lungs and through the skin and is absorbed little by the gastrointestinal tract [27]. The interaction between Hg and the skin has relationships with affinity between Hg and the sulphide group of proteins [56,57]. In a study conducted by Schneider et al. (2022) that evaluated skin samples of Natator depressus collected off the coast of Australia, which present likely eating behaviors, Hg concentrations were 0.017 mg·kg^−1^; in this study, the average concentrations of HgT in *P. expansa* skin were 0.1045 mg·kg^−1^ [58].

#### 3.2.3. THg Bioaccumulation

Toxic metals entering the river body would be absorbed in sediments and migrate to the sediment and biota through a biological and chemical process [59,60], so researchers began to pay more attention to pollution by toxic metals [61,62].

The bioaccumulation of toxic metals by reptiles results from surface contact with the water and food chain [47,63,64], and the concentrations depend on the environmental levels of trace metals in the habitat, since they are absorbed and stored in the tissues [65]. Researchers have shown that the mercury concentrations in the muscles of freshwater animals varies widely depending on the capture location [59,66].

Captive breeding possibly contributed to reduced mercury concentrations compared to the reported data [57]. The researchers evaluated the THg concentrations in the muscles and keratin samples from the carapaces of free-living *P. expansa* in the Amazonas region and found THg concentrations in the muscle of 0.1 mg·kg^−1^ and in the carapaces of 2.8 mg·kg^−1^. However, the diet and food consumption were human-controlled, thus interfering with the Hg kinetics.

In addition, the muscle THg concentration is related to eating habits, which interfere with the biomagnification of this trace element. Previous research regarding the THg concentrations in Amazon chelonian muscles from different species found concentrations of 0.106 mg·kg^−1^ in *P. expansa,* corroborating the results of the present research [51].

*C*. *fimbriatus*, a predatory carnivorous animal, is adapted for collecting fish and other animals in the aquatic environment [67] via a well-developed suction system and head movements toward the prey [68]. In contrast, when raised in captivity, young *P. expansa* prefer meat and fish [53], which may explain the higher average THg concentrations found in *C. fimbriatus* specimens compared to *P. expansa*.

Different Amazonian reptiles have been used to monitor THg bioaccumulation. Researchers evaluated this trace metal in various *C. crocodilus* tissues, this being an alligator from the river Purus (Amazonas state), and found a THg concentration of 0.389 mg·kg^−1^ [69]. Correia et al. (2014) evaluated THg in alligators from the Mamirauá reserve (Tefé, Amazonas State) and found THg concentrations of 0.407 mg·kg^−1^; these animals showed predatory eating behavior [70]. Thus, the THg concentrations represent the biomagnification principle, which is associated with the consumption of fish, chelonians, small mammals and birds.

#### 3.2.4. Human Health Risk Assessment

Although the established Provisional Tolerable Weekly Intake (PTWI) limits [12,19,20] suggest safe mercury intake values, researchers and global health organizations highlight that this metal is dangerous at any concentration due to its bioaccumulation characteristics, and thus there is no healthy mercury level for humans [35]. Bioavailability can be affected by controlling the biotic and abiotic factors. The most critical tool that can be used to reduce mercury in the environment and in animals is the prevention of pollution, which is considered a long-term goal [71]. In the state of Amazon, Balbina village (Vila Balbina) is highly influenced by mining and damming, both of which are anthropogenic activities on the River Uatumã causing environmental impacts and interfering with the Hg biological cycle [28].

The maximum tolerable limit of 0.5 mg MeHg·kg^−1^ for fish (except for predatory fish) [22] is established for seafood based on a weekly consumption. The Brazilian regulatory agencies also set the acceptable THg level for non-predatory fish, crustaceans, mollusks, cephalopods and bivalve mollusks at 0.5 mg·kg^−1^ and at 1.0 mg·kg^−1^ for predatory fish [21]. Although regulatory agencies have not established a THg limit for chelonians, it is important to note that, when comparing to the limit set for non-predatory fish, the THg concentration detected in the muscle of *P. expansa* (0.1045 mg·kg^−1^) remained below the legal thresholds. However, it should be highlighted that this study identified two (2/19) *P. expansa* with THg concentrations exceeding 0.5 mg·kg^−1^, which surpasses the levels established for non-predatory fish.

A Hazard Quotient (HQ) above 1 (HQ > 1) represents a potential health risk to consumers [20]. To determine the HQ for the consumption of *P. expansa*, it is crucial to consider the weekly consumption of the male categories. When considering the mean THg concentration in males and females for monthly or annual consumption, the HQ was <1 in all categories. Research carried out by Correia et al. (2014) [70] evaluated the HQ for alligator consumption by riverside populations and determined a value of 7.19, considering this quotient due to the higher Hg concentrations in these animals as compared to *P*. *expansa*, since they show predatory feeding behavior and are piscivores.

One of the characteristics of the consumption of turtles in the Amazon area is its seasonality, with the hunting of turtles and other animals for consumption being more intense in preserved and isolated areas [35]. In the present study, 18% (15/91) of the chelonian meat consumers interviewed considered it rare and occasional, only occurring in months with greater availability. Turtle consumption is taboo among the Amazonian population due to preservation practices of these species and constant supervision by environmental agencies [26].

The proximity and coexistence of Vila Balbina residents with environmental control bodies such as the Instituto Chico Mendes de Conservação da Biodiversidade (ICMBio) and the Instituto Brasileiro do Meio Ambiente e dos Recursos Naturais Renováveis (IBAMA) may have influenced the responses regarding the consumption of chelonian in the questionnaire administered to residents.

Furthermore, considering the accessibility of the study area, which is close to the state capital, Manaus (distance 107 km), it is reasonable to assume that consumption levels may vary in more isolated regions, such as indigenous, riverside and rural communities, where access to alternative sources of animal protein production is limited [26,71].

Although in this research only a small percentage of individuals reported consuming turtle meat and eggs, the true consumption might be higher. Female specimens are more consumed since they are easier to capture, their meat is more appreciated and the yield is higher [72]. The increased ease of capturing female *P. expansa* is associated with the time of egg-laying when these animals are more vulnerable. During this period, they leave the aquatic environment and deposit their eggs in nests on riverbeds. They are often captured, and the collected eggs, aside from being valued for their flavor, also hold medicinal purposes in ethnoknowledge, with claims of aphrodisiac properties.

Little information is available on the risk evaluation associated with chelonian consumption. Studies emphasized that total mercury concentrations in river fish from Malaysia and Ghana were influenced by the habitat, feeding habits, species and body length, and the increased HQ was due to the nearby gold mining area [73]. Brazilian researchers have reported HQ values ranging from 1.5 to 28.5 in the river Amazon and concluded that the elevated risk was associated with artisanal gold mining areas, where people also catch fish [1]. They suggested population risks at almost the sampling points due to consumption. It appears that there is considerable contamination of living beings that share the same environment with turtles due to their consumption without considering food safety [74]. Hence, determining the allowable limit for the consumption of these chelonians (daily, weekly or monthly) is crucial to preserve public health.

## 4. Conclusions

*P*. *expansa* chelonians, kept in captivity in Vila Balbina (Balbina village) at the mouth of the River Uatumã, accumulated THg in the muscles, skin and brain, with higher concentrations in the skin and muscle. Regarding gender differences, the males showed higher THg concentrations than the females. The concentration found in the *P. expansa* muscles were generally lower than the limits established for fish consumption.

The mercury content is mainly associated with animals’ diets and is considered to accumulate in organisms. It is essential to notice that depending on the consumption characteristics, the frequency and animals’ gender may be a risk to public health. Although the THg concentrations are below the acute metal toxicity level, the bioaccumulation characteristic deserves attention because of its long-term effects on human health.

Research is essential to assess the risk of Hg exposure resulting from the consumption of game meat, including chelonians meat, which holds a significant place in the diets of indigenous and riverside communities. It is imperative to evaluate mercury contamination in both wild-caught game animals and those raised in captivity through commercial breeding for meat production. Furthermore, analyzing a range of game animals from diverse origins and studying the consumption habits of different populations will contribute to understanding the significance of these meats as a potential source of Hg exposure.

A comprehensive study on this trace element in the Amazonian area is recommended to determine allowable consumption limits and identify the location most involved with contaminants. There are clear benefits and risks from the consumption of fishery products, so the population should be provided with enough information to maximize the positive health benefits while minimizing the risks from contaminants. General and scientific knowledge is scarce, reinforcing the importance of developing communication tools involving scientists, health professionals, regulators and the general public.

In addition to the need for further studies on the presence of Hg in the meat of chelonians from different species, locations and sexes, it is also important to investigate the Hg content in eggs. Furthermore, research on the consumption of these products is necessary to conduct risk assessments. In this study, *P. expansa* eggs were not collected and analyzed due to the seasonal nature of egg-laying, and the availability of this material was limited in the researched area.

Efforts should be established to address illegal mining in Brazil, especially in the Amazon region. These efforts include establishing controls in unlawful mining areas, increased surveillance of borders and indigenous territories, and strengthening environmental enforcement agencies such as ICMBio and IBAMA. Additionally, there is a need for continuous monitoring of mercury concentrations in fish and other food sources, along with regular risk assessments, to prevent the consumption of foods with high mercury levels.

## Figures and Tables

**Figure 1 ijerph-20-06489-f001:**
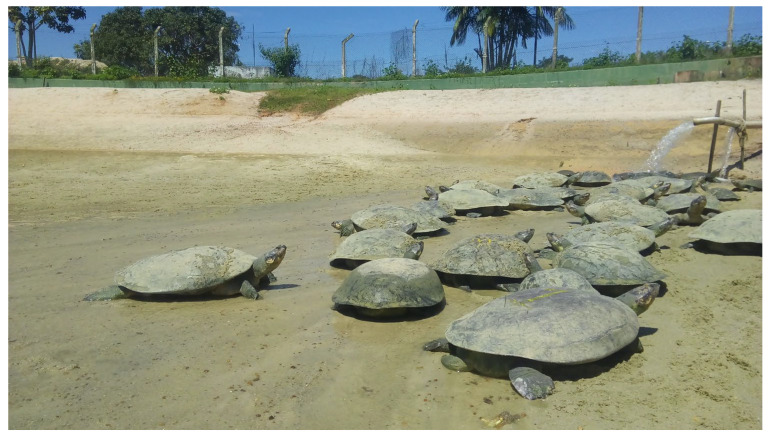
Amazonian giant river turtle (*Podocnemis expansa*) in the herd of the Centro de Preservação e Pesquisa de Quelônios Aquáticos (CPPQA), Vila Balbina, Presidente Figueiredo municipality, Amazonas state, Brazil.

**Figure 2 ijerph-20-06489-f002:**
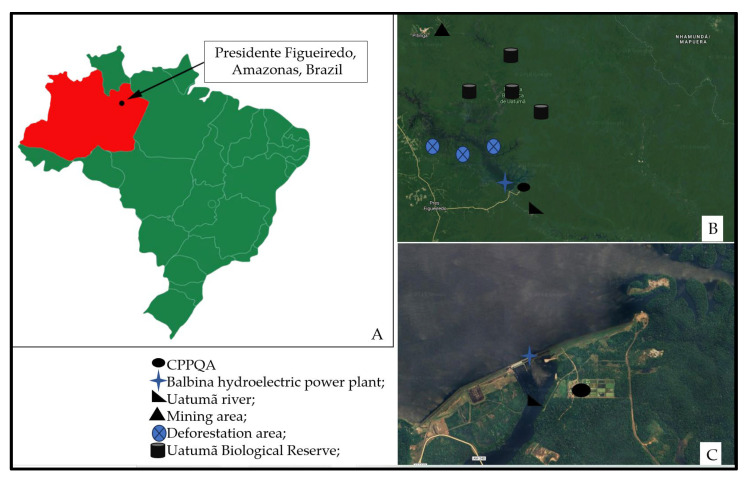
Location of sample collection in Centro de Preservação e Pesquisa de Quelônios Aquáticos (CPPQA), Vila Balbina, Presidente Figueiredo municipality, Amazonas state, Brazil. (**A**): Location in Brazil; Red: Amazonas state; Arrow and dot black: Presidente Figueiredo municipality; (**B**): 200 miles; (**C**): 200 m.

**Table 1 ijerph-20-06489-t001:** Mean, standard deviation, minimum and maximum of total mercury (THg) concentrations in different tissues of Amazonian giant river turtle (*Podocnemis expansa)*.

Tissues	Gender	*n*	Carapace (cm)	THg (mg·kg^−1^)	
				Mean ± SD	Min.–Max.
Skin	F	12	75.00	0.0817 ± 0.0165 ^a^	ND–0.0406
	M	16	46.64	0.1215 ± 0.1401 ^a^	ND–0.5457
Total Skin		28		0.1044 ± 0.2160 ^a^	ND–0.5457
Muscle	F	5	74.83	0.0152 ± 0.1019 ^b^	ND–0.6929
	M	14	47.18	0.1427 ± 0.1132 ^a^	ND–0.5677
Total Muscle		19		0.1092 ± 0.1710 ^a^	ND–0.6929
Brain *	F	2	77.00	0.0600 ± 0.0009	0.0591–0.0610

ND: not detected; *n* = number of samples; * Considering scarce samples statistics were not applied. ^a,b^ Means followed by the same letter in the same column do not statistically differ by the Tuckey test at 5% probability (*p* < 0.05).

**Table 2 ijerph-20-06489-t002:** Human health risk assessment for the meat of Amazonian giant river turtle (*Podocnemis expansa*) by gender and ingestion frequency.

Frequency Consumption	Gender of *P. expansa*	Daily Intake (I) (mg/kg/day)	Hazard Quotient (HQ)
Annually	Male Female Mean	1.85 × 10^−5^ 2.00 × 10^−6^ 1.41 × 10^−5^	0.061 0.006 0.047
Monthly	Male Female Mean	2.21 × 10^−4^ 2.36 × 10^−5^ 1.70 × 10^−4^	0.739 0.078 0.565
Weekly	Male Female Mean	9.61 × 10^−4^ 1.02 × 10^−4^ 7.35 × 10^−4^	3.204 0.341 2.452

## Data Availability

All data generated during analyzed this study are included in this article.

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
