# Peer review of "Total Mercury Content in the Tissues of Freshwater Chelonium (Podocnemis expansa) and a Human Health Risk Assessment for the Amazon Population in Brazil"

_ijerph, 2023, doi:10.3390/ijerph20156489_

Round 1

Reviewer 1 Report

Dear Authors,

Its my pleasure to be reviewing the research and suggesting comments aimed at betterment of the article. I find the research interesting, but I will like put forward few points aimed at improvement of the manuscript .

1. Abstract-- Line number 13 "Researchers are highly aware of the silent, negative effects caused by mercury pollution in gold mining areas."-- try to avoid indefinite word choices such as 'highly aware'. 

2. Line- 43-44: 'less visible than others' is not a scientific writing. This need to be chaged.

3. Authors should give examples of Hg pollution at Water-soil-fauna matrix, their use as food, from different regions of the globe to substantiate the argument and to note the works already done in the domain. Some such works are, https://www.tandfonline.com/doi/abs/10.1080/10807039.2017.1415131, https://www.sciencedirect.com/science/article/abs/pii/S0013935100940350 (amazon region specific), https://www.sciencedirect.com/science/article/abs/pii/0048969795049096. Section 2.3: Note the methodology of sampling in detail and how authors have got the data on consumption pattern. 

7. section 2.9: Is the data set follows the statistical assumptions required for a ANOVA analysis? Authors need to justify the use of the statistics.

8. Human health risk analysis should be done using the USEPA guidelines (https://www.tandfonline.com/doi/abs/10.1080/10807039.2017.1415131) before drawing any conclusion.

Otherwise, I find this article interesting and novel. 

I will suggest authors to get the manuscript checked by a native english speaker for language consistency. 

Author Response

Att.,

Reviewer 2 Report

The authors measured mercury concentrations in Podocnemis expansa in Brazil and discussed the impacts on local populations and origin of mercury. This study assessed the risk of mercury in turtles, of which information was little. The results are important, interesting and worthy of publication.

Followings are the points that should be modified for the publication.

Overall, It is better to unify the names of the turtle of target in this study. For example, the authors mentioned such as “chelonian samples”, “samples of Giant South American freshwater turtles (P. expansa)” in P2, “Amazonian giant river turtle (Podocnemis expansa)”in Table 1, “P. expansa” in L196, “The Giant South American river turtles (P. expansa” in L212. If the authors would like to use abbreviation, then it should be also unified after mentioning the formal name.

P1, L28: Isn't WHO currently warning the Amazon population of the hazard of metal? If any warning is given, it should be clearly stated in the text how far the warning is and what is lacking. If WHO haven't warned at all, you should mention about it in the text as the important future task, too. 

P1, L37: Hg would not be an “essential” element.

P2, L55: The period should be black. 

P2, L75-79: Since it is related to the origin of Hg and is an important text, it needs to be described in more detail. Also, is this Hg methylmercury, inorganic, or is it known? Which type of mercury concentration to measure?

P2, L87: It should be mentioned first whether "The chelonian sample" is a live turtle, a dead turtle, or different from part to part.

P2, L88: Figure 1 was not in the manuscript.

P2-3, 2.2: How were the turtles treated after they were biopsied?

P3, 2.3: You should explain in detail how to do the questionnaire. Interview methods, etc.

P3, L135: Information such as the degree of agreement with the CRM concentration (ex. The) percentage of deviation) is required.

P4, L168: “WHO (2010)” was not in reference. Also, what RfD? Oral?

Is it mercury, not methylmercury, that accumulates in turtles? It is also necessary to be mentioned in the text.

P4, L183: Regarding “THg”, it is necessary to write the official name at the first appearance.

P5, L204-205: R2 = 0.43 is written as if there was not much correlation observed, but R2 = 0.43 is not significantly different from R2 = 0.5, and is similarly defined as having a relatively strong correlation. Please confirm the definition and discuss.

P5, L211-212: Is there a difference in the rate of Hg accumulation between the two species of turtle? Also, although this study is only for P. expansa, what would be the estimated overall Hg exposure if the population were eating several species of turtle?

P5, L212-217: Although eggs were mentioned in the text, there was no discussion of risk assessment for eggs or things to be related to eggs in the text. Also, what is the significance of eating turtles during the egg-laying month?

P6, L236: There is extra space in the text.

P6, L239-244: Is there any previous research showing that Hg is likely to accumulate in stalks and shoots that males tend to eat more?

P6, L244-246: Since the authors did not evaluate eggs, this quote (Frossard et al., 2021) seems to be abrupt.

P6, 249-251: Earlier in this section, the authors refered to Hg kinetics, but following discussion does not lead to this conclusion (Especially about the “anthropic action”). Related contents and citation are needed, if you would like to conclude with “Anthropic action” on Hg in this section.

P6, L255-261: It would be better to connect to the first discussion in 3.1.2, such as estimating human exposure considering the differences in Hg concentration between turtle species and eaten ration of each species.

P7, L301-303: The author mentioned about the free-living turtles of which “the diet and food consumption were human-controlled, thus interfering with the Hg kinetics.”, but the Hg concentration in muscle was 0.1 mg.kg-1. This value is the same as that of completely free-living P. expansa which the authors measured. Then, it cannot be say that human-control can interfere with the Hg kinetics?

P8, L324: Regarding “PTWI”, it is necessary to write the official name at the first appearance.

P8, L341: “Although” appears twice in the same sentence. What is “two turtles presented”? More explanations are needed about this two turtles.

P8, L356-358: What is the reason that the author considered “the true consumption might be higher”? Since eating turtles is taboo, the people who answered the survey didn't tell the truth?

P8, L361: If you write as “The authors”, it would be easy to regard “the authors” as the authors of this paper.

P8, L364: In the reference, year of publication was written as “2014”.

P9, L379-390: There are many generalities. More contents of this study should be reflected in “Conclusions”, such as the bioaccumulation which was discussed in this study.

No major issues were found. Please see the comments.

Author Response

Att., 

Reviewer 3 Report

Mercury pollution is a significant environmental and public health issue in Brazil due to the expansion of gold mining activities, and therefore this study is indeed relevant. Furthermore, most studies investigating mercury contamination in animals analyze fish. The fact that this study analyzed turtles (a typical bushmeat practice in the Amazon region) makes this work even more relevant. However, I would like authors to consider the following points:

 - Abstract: briefly mention the place (i.e., city, state, country) where the analyzed turtles were sampled. Also, cite Podocnemis expansa in full for the first time this scientific name appears in the abstract.

 - Introduction/discussion: although the consumption of turtles is a cultural practice in the Amazon region, making up an important part of the diet of the region's population, it is important to mention that bushmeat consumption triggers a series of problems, from conservation issues to infectious disease risk. Take into account the information from this article: An Acad Bras Cienc. 2020; 92(1):e20191375. doi: 10.1590/0001-3765202020191375 Although this is not the main focus of the study, I consider it important to mention in the introduction or discussion section the problems involved with the consumption of turtles in addition to mercury contamination. This is a neglected issue in Brazil and deserves more attention (the authors only mention conservation issues very briefly in lines 354-356 of the discussion section).

- Introduction: environmental issues are disconnected from the real, political world. Consider mentioning the effects of Bolsonaro’s administration on the huge expansion of illegal mining in the Amazon region.

- 2. Materials and methods, 2.1. Experimental area: Figure 1 is mentioned in the text but is not shown in the manuscript. Please correct it.

- 2. Materials and methods: please include a picture of Podocnemis expansa.

- 2.3. Chelonian consumption data: please describe more detailed information/criteria concerning the selection of individuals included in the interviews.

- Discussion, line 234: correct “P. Unifilis” to “P. unifilis”.

- Conclusion: I strongly suggest that the authors cite some actions to curb gold mining activities and related mercury pollution in the Brazilian Amazon. 

English is of sufficient quality for scientific publication purposes.

Author Response

Att., 

Round 2

Reviewer 1 Report

The manuscript can be accepted in the current form.

Could improve.

Author Response

Dear Reviewer 1,

We greatly appreciate your considerations and review of our article, they were fundamental for improving our work.

Reviewer 2 Report

Thank you for your thoughtful revision. Most of them were well revised, but there are still some points to be revised as follows.

1. Texts in L122-129 were written in italic type. Please modified the form. Also the title of "2.5." is included in those text.

2. Please also refer in the text what you responded in question  (8), (14), (15), (21), (24). 

I also recommend to write the contents what you write in "the response of (14)" in "Conclusions" as limitations of this research and future prospects for this field.

3. In the response of (13), you wrote "an R² value of 0.43 is not significantly different from an R² value of 0.5, and both indicate a relatively strong correlation".

However, you wrote in the text L228-230, "Furthermore, the correlation between tissues (skin and muscle) was positive for males (R²=0.50), but a similar correlation was not observed for females (R² = 0.43)."

It seems that you wrote the opposite of the response of (13) in the text L228-230.

4. In response to (17), reading paragraphs L258-269, it was stated that males and females have preferences, and the text also states that there was a difference in the amount of mercury accumulated between them. Is it possible to point out something from previous research as to why the amount of accumulated mercury in them differed? If it is difficult to answer this question now, you should write reason. Because your results seem to relate for this discussion, so the consideration for this will attract the great interests from readers.

5. There is a half bracket in L270. Please delete it.

Thank you for your revision again. 

Author Response

Dear Reviewer 2, 

Reviewer 3 Report

The authors responded to my comments satisfactorily.

Author Response

Dear Reviewer 3, 

We greatly appreciate your considerations and review of our article, they were fundamental for improving our work.

Round 3

Reviewer 2 Report

Thank you for your revise. Please make revisions only for the following points.

1.Regarding Response 3, (L250-251), "no difference was observed" is too strong expression and also different from the fact. First you mentioned that males have correlation, but females don't have correlation, so I pointed out that both have correlation and they don't have significant difference, which means that "they still have a slight difference".

"No difference" and "slight (or small) difference" have different meaning.

Please rewrite the sentences "no difference was observed" to "a slight difference was observed" or "a large difference was not observed".

2. L284: There is an extra space in the text. Please delete it.

3. L404: "Km" should be "km".

After revising above points, I agree on the publish. Thank you.

Author Response

Dear Reviewer 2, 

Cordially
